# Slope Unit Maker (SUMak): An efficient and parameter-free algorithm for delineating slope units to improve landslide modeling

Jacob B. Woodard[1], Benjamin B. Mirus[1], Nathan J. Wood[2], Kate E. Allstadt[1], Benjamin A. Leshchinsky[3], Matthew M. Crawford[4]

[1] U.S. Geological Survey, Geologic Hazards Science Center, Golden, CO, USA
[2] U.S. Geological Survey, Western Geographic Science Center, Portland, OR, USA
[3] Department of Forest Engineering, Resources and Management, Oregon State University, Corvallis, OR, USA
[4] Kentucky Geological Survey, University of Kentucky, Lexington, KY, USA

*Correspondence to:* Jacob Woodard (jwoodard@USGS.gov)




**Abstract.**

Slope units are terrain partitions bounded by drainage and divide lines. In landslide modeling, including susceptibility modeling and event-specific modeling of landslide occurrence, slope units provide several advantages over gridded units, such as better capturing terrain geometry, improved incorporation of geospatial landslide-occurrence data in different formats (e.g., point and polygon), and better accommodating the varying data accuracy and precision in landslide inventories. However, the use of slope units in regional (>100 km$^2$) landslide studies

remains limited due, in part, to the large computational costs and/or poor reproducibility with current delineation methods. We introduce a computationally efficient algorithm for the parameter-free delineation of slope units that leverages tools from within TauDEM and GRASS, using an R interface. The algorithm uses geomorphic laws to define the appropriate scaling of the slope units representative of hillslope processes, avoiding the often ambiguous determination of slope unit size. We then demonstrate how slope units enable more robust regional-scale landslide

susceptibility and event-specific landslide occurrence maps.

**Short summary**

Dividing landscapes into hillslopes greatly improves predictions of landslide potential across landscapes but their scaling is often arbitrarily set and can require significant computing power to delineate. Here, we present a new computer program that can efficiently divide landscapes into meaningful slope units scaled to best capture landslide

processes. The results of this work will allow an improved understanding of landslide potential and can help reduce the impacts of landslides worldwide.

**1 Introduction**

Landslides cause substantial losses of life, infrastructure, and property every year across the world (Froude and Petley, 2018). One of the most common tools for mitigating these losses is landslide-susceptibility mapping, which

provides information on the spatial patterns and likelihood of landslide occurrence. Data-driven statistical models are typically used for creating these maps due to their computational efficiency and the relative availability of data needed to develop and deploy these models (van Westen et al., 2008). Statistical models analyze the spatial distribution of known landslides in relation to local terrain conditions (e.g., slope, curvature, aspect), and other areas with similar conditions are identified as being susceptible to landslides. In essence, the models identify features in

the terrain similar to known landslides as a measure of landslide susceptibility. As such, the quality of the landslide inventory used to develop the susceptibility model is paramount for creating reliable maps. However, inventories with accurate information on landslide positioning, extent, triggering mechanism, and type are unavailable in many parts of the world. More often, if an inventory exists at all, it consists of a compilation of landslide data collected at different scales, times, accuracies, and formats (e.g., polygons or points) with limited information on the landslide

type or triggering mechanism (Mirus et al., 2020).

Another tool used to mitigate losses associated with landslides are near real-time or forecasted landslide occurrence models (Nowicki Jessee et al., 2018; Nowicki et al., 2014; Tanyas et al., 2019; Kirschbaum and Stanley, 2018). Rather than characterizing the potential of landslide existence from static terrain conditions, these models include a dynamic input designed to characterize landslide potential from a particular forcing event. For example, Tanyas et

al. (2019), analyzed the static terrain conditions and dynamic ground motion metrics (e.g., peak ground velocity) from 25 earthquake-induced landslide-event inventories from across the world to create a landslide model that can estimate the distribution of landslides during an earthquake. Herein, we will refer to this model type as a landslide occurrence model. Like susceptibility models, landslide occurrence models often suffer from imperfect and heterogeneous landslide data. Thus, a common problem in the landslide community is determining an effective way

of assessing landslide susceptibility and/or occurrence, despite the imperfect data available for model development.

The foundation of any landslide map (susceptibility and occurrence) is the mapping unit used to subdivide the terrain for landslide analysis. Grid cells (pixels) are the most used mapping unit, constituting about 86% of all publications on landslide susceptibility as of 2018 (Reichenbach et al., 2018). This is due largely to their ease in processing. However, grid-based mapping units have several major drawbacks. First, the grid cells have no physical

relationship to landslide processes. Landslides occur at various spatial scales and manifest a large range of footprints

not appropriately captured by grid cells. Second, variable scales of data that describe the local terrain conditions used to develop landslide models (i.e., predictors or covariates) can lead to model biases. For example, the size of the grid cell can have major effects on the output of the landslide model (Chang et al., 2019; Guzzetti et al., 1999; Catani et al., 2013). To mitigate these effects, some researchers suggest creating multiple models at different

resolutions (e.g., Guzzetti et al., 1999). Third, landslide inventories are often mapped using a mix of formats (i.e., polygon and points). This requires modelers to standardize the data in some way (Zêzere et al., 2017; Jacobs et al., 2020; Süzen and Doyuran, 2004; Zhu et al., 2017; Tanyas et al., 2019). For regional-scale (>100 km$^2$) models that use high-resolution (<100 m) rasters, this standardization is often implemented by sampling a single representative cell from within each landslide polygon (Qi et al., 2010; Gorum et al., 2011; Xu et al., 2014; Oliveira et al., 2015).

Alternatively, some studies use lower resolution rasters (>100 m) and sampling all the cells that touch a landslide polygon or point (e.g., Nowicki et al., 2014).

Slope units alleviate many of the problems of grid mapping units and are based on drainage and divide lines that effectively segregate the terrain according to the hillslope processes that shaped it (Carrara, 1983; Guzzetti et al., 1999). First, the slope units' relationship with the natural terrain allows modelers to use an array of statistics of the

predictors inside of the mapping unit (e.g., max, min, standard deviation). Second, the amalgamation of grid cells to create a slope unit provides a natural subset of the terrain that reduces the need for multiple raster resolutions for the susceptibility analysis (Jacobs et al., 2020). Third, slope units provide an alternative solution for the incorporation of landslide data in different formats. In contrast to the common grid-based standardization procedures, slope units allow modelers to study the characteristics of the whole hillslope(s) that experienced a landslide. Fourth, slope units

are less sensitive to the effects of inaccurate landslide locations (Jacobs et al., 2020). Finally, although the use of slope units requires more processing at the beginning of the analysis, the limited number of mapping units enables the use of input data from every mapping unit, even over large regions. The representation of every mapping unit in the study area prevents the potential of sampling bias common when using grid mapping units (e.g., Oommen et al., 2011; Petschko et al., 2013).

Recognition of the advantages of slope units has led to many different methods for delineating them. However, the disadvantages of these methods include inhibiting computational costs, time-intensive manual cleaning and/or delineation, or indeterminate parameterizations that control the slope units' scaling. For example, the most rudimentary method for creating slope units is using watersheds to draw their boundaries (Carrara, 1988). A drawback of this approach is that the sizes of the slope units are determined by the user and the cleaning of artifacts

which occur during the watershed delineation process can be highly labor intensive and difficult to reproduce. Computer-vision techniques (e.g., landform classification) have also been used to delineate slope units (Luo and Liu, 2018; Martinello et al., 2022; Zhao et al., 2012; Cheng and Zhou, 2018) which overcome the reproducibility and labor issues of the manual delineation method. However, the scale of the slope units is still often arbitrarily set. The algorithm *r.slopeunits* developed by Alvioli et al. (2020, 2016) uses watershed delineations whose shape and

dimensions are determined by the user or an iterative optimization procedure (i.e., a parameter sweep) that evaluates the algorithm's outputs while using different input parameter values (see Alvioli et al., 2020, for details). Although the algorithm can avoid manual parameter assignments (i.e., parameter free), the computational expense of the parameter sweep can be prohibitive for large areas. For example, Alvioli et al., (2020) summarizes a three-month process to delineate slope units based on a 25 m digital elevation model (DEM) for the country of Italy while

omitting the flat regions (~24% of the total area) using a 64-core machine with 320 GB of memory. Additionally, the optimization procedure required for the parameter-free delineation of slope units is not openly available. The limitations of all the current slope unit delineation methods prevents the widespread use of slope units in susceptibility modeling.

The scaling of slope units should not be arbitrarily set to avoid the modifiable areal unit problem (MAUP)

(Openshaw and Taylor, 1983; Buzzelli, 2020; Goodchild, 2011). The MAUP occurs when the cartographic representation of data varies significantly by the scale of the mapping unit used to represent the data. MAUP is a challenging issue to overcome; however, determining a scale of the slope units so that they effectively capture the hillslope processes that lead to landslides can greatly mitigate the negative effects of the MAUP (Buzzelli, 2020). Alvioli et al. (2020) recognized this challenge, which motivated the development of their custom optimization

procedure. Importantly, the optimal scale for capturing hillslope processes is spatially variant. Thus, the ideal scaling of slope units should adjust to the local topography.

The objective of this paper is to introduce Slope Unit Maker (SUMak), an open-source, slope-unit delineation tool that is computationally efficient and parameter-free and to demonstrate how slope-unit based landslide maps are generally a better mapping unit for regional (>100 km$^2$) landslide analysis. SUMak leverages the watershed

optimization algorithm available in the software package 'Terrain Analysis Using Digital Elevation Models' (TauDEM) (Tarboton, 2015) to determine the optimal scale of the watersheds for capturing hillslope processes. This optimization avoids the computationally inefficient parameter sweeps required by other parameter-free algorithms, making it markedly faster. To demonstrate the utility of SUMak, we divide this manuscript into two parts: 1) an explanation and demonstration of our slope unit delineation algorithm, 2) an example of how slope units are

generally a better mapping unit for regional landslide modeling due to the larger mapping units that align with the local terrain. In part two, we first show that slope units provide a conservative means of displaying the nebulous susceptibility model output caused by imprecise input data (e.g., no time component, imprecise locations, and/or variable formats). We do this by comparing landslide susceptibility map outputs from grid and slope unit-based maps in two watersheds in the state of Oregon (U.S.) which have inventory data mapped at a range of scales and

formats. Next, we demonstrate the advantages of slope units for assessing event-based landslide occurrence using a landslide catalog from Hurricane Maria over the island of Puerto Rico (Hughes et al., 2019). Landslide models are developed using logistic regression and XGBoost machine learning algorithms.

**2 Methods and Data**

**2.1 Slope unit delineation**

To efficiently map slope units over a given terrain, we adapt tools from the software TauDEM (Tarboton, 2015) which determine the scale where the topography transitions from fluvial to hillslope processes using the constant drop law (Figure S1). The constant drop law states that the average drop in elevation along Strahler stream orders (Strahler, 1957) is constant (i.e., independent of order) at scales, or aerial extents, of the terrain controlled by fluvial processes. At sufficiently small scales, the constant drop law does not hold, indicating that hillslope processes are

controlling the terrain morphology. The scale at which the constant drop law breaks is determined by applying a series of flow accumulation thresholds to the input DEM and finding the threshold where the mean stream drop of the first order streams is statistically different from the higher order streams, using a T-test (Davis, 2002). The stream accumulation threshold just below where the law breaks is then used to delineate the largest watersheds that capture the hillslope processes of that terrain. This scaling law is independent of the raster resolution (Tarboton et

al., 1991; Tarboton, 1989) and has been used extensively in the field of fluvial geomorphology. We further process these optimally scaled watersheds by splitting them by the longest flow path within the watershed using GRASS (GRASS Development Team, 2020). Thus, the watersheds essentially become what would be objectively recognized as a slope. We argue that basing the scaling of slope units used for landslide analysis on established geomorphic laws provides the best justification for their appropriate sizing and odds of mitigating the negative effects of the

MAUP.

If the domain of interest has significant variation in topography, TauDEM may choose a threshold that doesn't adequately characterize every area within the domain. Thus, SUMak provides different options for subdividing the domain in preparation for the application of the slope unit optimization procedure described above. We refer to these preliminary subdivisions as intermediate watersheds. Intermediate watersheds must be small enough to limit the

variation in topography but large enough to avoid significantly reducing computational efficiency. While experimenting with different watershed dimensions on the topographically diverse regions of Sicily, Puerto Rico, and the Umpqua and Calapooia watersheds, we found an accumulation threshold of ~100 km$^2$ to adequately strike this balance. This threshold can be adjusted to meet the user's needs, or SUMak has an option to input predetermined intermediate watersheds. After appropriate intermediate watersheds are created, the algorithm runs

the rest of the processing steps individually for each intermediate watershed in parallel as detailed in Text S1 and the online repository (Woodard, 2023).

**2.2 Susceptibility maps**

Several papers have evaluated the relative effectiveness of slope units over grid mapping units in statistical landslide susceptibility models (Jacobs et al., 2020; Steger et al., 2017; Zêzere et al., 2017; Van Den Eeckhaut et al., 2009; Martinello et al., 2022). However, none of these studies has thoroughly evaluated the effectiveness of slope units for better visualizing the imprecise susceptibility model outputs caused by inconsistent input data or their advantages in displaying near real-time or forecasted landslide occurrence maps. To demonstrate these benefits, we use the Middle Umpqua and Calapooia 10-digit hydrologic unit code (HUC) watersheds (U.S. Geological Survey, 2004) in the State of Oregon (U.S.) and the island of Puerto Rico which have areas of 257 km$^2$, 743 km$^2$, and 8,870 km$^2$, respectively. Each area's landslide catalog includes an assortment of landslide types (slumps, debris flows, rockfalls, deep-seated landslides, and others), which are not differentiated in this study. The landslide data from the Oregon were collected over decades using a combination of 1-m DEM data and its derivatives, geologic maps, orthophotos, aerial photography, and field reconnaissance and consists of both point and polygon data (Burns and Madin, 2009). The Oregon landslide catalogs contains no temporal constraints on landslide occurrence. The Umpqua dataset contains 941 points and 3213 polygons, while the Calapooia dataset contains 33 points and 456 polygons. In this dataset, polygons cover the extent of the landslide affected area while points are placed at the centroid of the landslide affected areas. All data were reviewed for accuracy after their initial mapping. The areas of the individual landslides mapped using polygons are highly variable, spanning 30-4.4×10$^6$ m$^2$ and 1500 - 1.88x10$^7$ m$^2$ in Umpqua and Calapooia, respectively. This data variability can lead to problems when using grid mapping units because the landslide data is standardized to a consistent format for the creation of the landslide susceptibility models. The Puerto Rico landslide dataset consists of 71,431 point locations of the centers of landslide headscarps that occurred during Hurricane Maria on September 20-21, 2017 (Hughes et al., 2019). Headscarps were manually identified using high-resolution (15-50 cm), post-event imagery and quality checked by three experienced supervisors. Importantly, the output of the landslide models for Puerto Rico are not a susceptibility map, rather a landslide occurrence map. That is, the models output the probability of a landslide occurring during Hurricane Maria. This type of output is similar to the landslide models developed for near real-time or forecasted assessment of event-specific landslides (Nowicki Jessee et al., 2018; Nowicki et al., 2014; Tanyas et al., 2019; Kirschbaum and Stanley, 2018). Our example from Hurricane Maria is intended to show how event-specific model outputs might differ between slope unit and pixel-based assessments. Thus, the Oregon watersheds and Puerto Rico datasets are used to demonstrate the benefits of slope units when using inconsistent and event-based input data, respectively.

We evaluate four different methods of standardizing landslide polygons to points for grid-based susceptibility maps in the Oregon watersheds. Each method converts the polygons to points which are combined with the landslides originally mapped as points. The first method converts the landslide polygons into a single point at the highest elevation cell within the polygon using a 10 m DEM from the US Geological Survey's three-dimensional (3D) Elevation Program (3DEP) database (U.S. Geological Survey, 2019), which has a vertical root mean square error of 0.82 m (Stoker and Miller, 2022). In cases where there are multiple points, the highest elevation cell with the highest slope is selected. This sampling method is designed to capture the attributes nearest the landslide scarp and the conditions that led to failure (Zêzere et al., 2017; Süzen and Doyuran, 2004; Jacobs et al., 2020). The second method follows the same procedure but is conducted using the same 10 m DEM resampled to 30 m resolution using a bilinear interpolation method. The coarser raster may better average the landslide characteristics compared to the finer-resolution rasters. Third, we sample multiple random points from the 10 m DEM within the polygons with a 200 m spacing, roughly halfway between the average radii of the landslide polygons from the two study sites (93 and 386 m for Umpqua and Calapooia, respectively). Each landslide polygon is guaranteed at least one point. Creating multiple points within the polygons allows us to capture some of the variability in the large landslides' measured attributes without eliminating the influence of landslides originally mapped as points. Using all the raster cells within the polygons would oversaturate the model with data from the landslide polygons and greatly reduce the influence of the landslides originally mapped as points due to their relative sparsity. Finally, we sample a point within each polygon at the median elevation value using the 10m DEM. In the case of multiple points per polygon, we select the point with the highest slope. This dataset is used to verify that the chosen statistics in the slope unit-based approach did not bias the results and to make the standardization more compatible with the Oregon point data. We refer to these four sampling methods as "10m", "30m", "10m_multi", and "10m_med", respectively. For the Puerto Rico dataset, we only use the "30m" sampling method as this dataset is used to demonstrate the use of slope units for event-based landslide inventories rather than inconsistent inventories. For all study sites, non-landslide data

are randomly sampled from areas outside the landslide polygons and points buffered with a radius derived from the average area of the landslide polygons within each study area. For Puerto Rico, this radius is set to a value between the two Oregon mean polygon radii (100m). If a landslide originally mapped as a point is within the boundaries of a landslide polygon, it is removed before standardization. The sampling ratio of landslide and non-landslide points is set to 1:1, following the most common practice (Petschko et al., 2013; Reichenbach et al., 2018). Table S1 shows the number of points for each study site and sampling method, respectively.

Slope units for the study sites are delineated using the same 10 m DEM as the grid-based approaches. We note that slope units can be delineated with coarser resolution elevation data with a loss in precision. The sampling scheme for the slope unit-based maps is simpler than the grid-based schemes. Each slope unit in the study area is set to be either a landslide sample or non-landslide sample dependent upon the intersection of a landslide point or polygon within that slope unit. We use an overlap threshold of 0.1% (i.e., at least 0.1% of the slope unit is covered by a

landslide polygon) for determining the positive presence of landslides within a given slope unit (Jacobs et al., 2020). Figures S2-S3 illustrate the slope units that contain landslides. In the Umpqua, Calapooia, and Puerto Rico study sites, 68%, 28%, and 4% of the slope units contained landslides, respectively. For the slope unit-based maps, we train two different models. The first uses only the median value of the predictor data within the slope unit and the other uses the median and standard deviation (SD) of the predictor data. To assure that the sampling ratio does not

bias the comparison between the slope unit and grid-based maps, we set the sampling ratio of landslide and non-landslide locations to 1:1 for the slope unit maps.

We created landslide susceptibility models using the logistic regression and XGBoost (Chen and Guestrin, 2016) machine learning algorithms. Logistic regression is the most commonly used algorithm for data-driven landslide susceptibility modeling (Reichenbach et al., 2018). It calculates the log odds ($\log(P/1 - P)$, where $P$ is the

probability), of a binary outcome given some predictor data ($x$) that describes the terrain. For $M$ input predictors, logistic regression is expressed as follows:

$$\log\left(\frac{P}{1 - P}\right) = \beta_o + \beta_1 x_1 + \beta_2 x_2 + \ldots + \beta_M x_M. \tag{2}$$

The input data's coefficients ($\beta$) are fit to the input data using a maximum likelihood criterion. XGBoost (https://xgboost.readthedocs.io/) uses a gradient boosting decision tree algorithm that increases in complexity until the lowest model residuals are reached (Chen and Guestrin, 2016). This algorithm is fast, easy to implement, and has

been shown to produce highly accurate susceptibility maps (Sahin, 2020). To increase the model accuracy while preventing overfitting, we optimize the 'max_depth', 'min_child_weight', 'subsample', 'gamma', and 'colsample_bytree' hyperparameters of XGBoost (see Text S2 for an explanation of these parameters) using a Bayesian cross-validation procedure. In short, these hyperparameters adjust how the model adapts to fit the training data. The Bayesian cross-validation procedure uses ten folds and ten iterations and assess the results from the

previous iterations to inform the next iteration of hyperparameters to use (Snoek et al., 2012). This procedure prevents the use of unwieldly grid searches and permits faster optimization of the model hyperparameters. For both algorithms, we limit the predictor variables to elevation, slope, aspect ($\phi$), roughness (standard deviation of the elevation using a 100 m square window), and curvature to illustrate the effectiveness of the different models using only widely available data. Aspect is measured using $\cos(\phi - 45°)$ to make it periodic and to account for variations

in solar heat flux (McCune and Keon, 2002). As the Puerto Rico landslide dataset has a known trigger, we also include root zone soil moisture estimates from NASA's Soil Moisture Active Passive (SMAP) mission on September 21, 2017. Bessette-Kirton et al. (2019) found the SMAP data to be a better predictor of landslide distributions from Hurricane Maria than other rainfall datasets. After the models are trained, we generated maps by applying the trained models to the entire study areas.


Importantly, the meaning of the models' output probability is different depending on the sampling methods used. The single-cell methods ('10m', '30m', '10m_med') measure the probability of a cell containing the high point (scarp) or center point of a landslide deposit recognized by the team(s) that compiled the landslide inventory. The multiple cell method ('10m_multi') is measuring the probability of a cell containing a landslide deposit recognized

by the team(s) that compiled the landslide inventory. Lastly, the slope-unit based maps measure the probability of a slope unit containing a landslide recognized by the team(s) that compiled the inventory. For the two Oregon

watersheds, the probability output of each method is used as a measure of landslide susceptibility. In contrast, the Puerto Rico maps output the probability of landslide occurrence during Hurricane Maria.

We measure the accuracy of the landslide models using the area under the curve (AUC) of the receiver operator characteristics (ROC) and the Brier score (Brier, 1950). The ROC curve compares the true positive rate against the false-positive rate at various discrimination thresholds (see Oommen et al., 2011 for an overview). If every landslide and non-landslide from the data is modeled correctly, the AUC values of the ROC curve will be 1.0. In contrast, AUC values near 0.5 suggest the model classification is equivalent to random guessing. Values from 0.5-0.6, 0.6-

0.7, 0.7-0.8, 0.8-0.9, and 0.9-1.0 can be classified as poor, average, good, very good, and excellent performance, respectively (Yesilnacar, 2005). The Brier score ($B$) measures the mean-square error between the model predictions (i.e., probability, $P$) and observations (binary variable of landslide presence, $O$):

$$B = \frac{1}{N}\sum_{i=1}^{N}(P_i - O_i)^2,$$ (3)

where $N$ is the number of observations (Brier, 1950). Thus, a $B$ value of zero suggests perfect model fit and a value

of one indicates perfect misfit. In contrast to AUC-ROC, the Brier score provides measure of the scale of the model fit and not just its ordering of landslide and non-landslide observations. Both metrics together provide a comprehensive evaluation of the model results. Following common practice (e.g., Molinaro et al., 2005), we use 70% of the data to perform a 10-fold cross-validation procedure with ten iterations to optimize the models parameters and obtain representative distributions of the ROC-AUC and Brier score metrics, while reserving 30% of

the data as a final test set. Model development and post-processing is conducted within R (R Core Team, 2022).

**3 Results**

**3.1 SUMak Slope Unit Delineation**

SUMak quickly delineates slope units over the three study areas while automatically adapting the scaling of the

slope units by the local terrain. Table 1 shows the time to delineate each of the study areas. Both Oregon watersheds were delineated in only a few minutes while the island of Puerto Rico took substantially longer. This is due to the larger area and the increased complexity of the delineating watersheds near coastlines where watersheds get increasingly small due to decreased accumulation areas. The adaptation of the slope unit sizes to the local topography is apparent in the slope unit maps (Figures S4, 1-2). For example, the Calapooia Watershed includes a mountainous and flat region (Figure 1). SUMak creates smaller slope units over the flat region compared to the

mountainous region to accommodate the difference in scale where hillslope processes occur (Figure S4).

**Table 1.** SUMak performance metrics.

| Location | Area (km²) | Coastline | DEM Resolution (m) | Compute Time (minutes) | Slope Unit Count | Time per area (seconds/km²) | Time per Slope unit (seconds) |
|---|---|---|---|---|---|---|---|
| Umpqua | 257 | No | 10 | 3.11 | 3841 | 0.7 | 0.05 |
| Calapooia | 743 | No | 10 | 9.97 | 6990 | 0.8 | 0.09 |
| Puerto Rico | 8870 | Yes | 10 | 383.28 | 140367 | 2.6 | 0.16 |

**3.2 Landslide map comparison**

Comparison of the final landslide maps to the distribution of landslide deposits highlights several differences between the grid and slope unit-based maps. The landslide inventories and examples of the grid sampling methods for the Oregon watersheds and Puerto Rico are in Figures 1 and 2, respectively. The slope units provide a division for landslides that enables the characterization of the entire slope(s) that experiences a failure (Figures 1c, d, 2). In contrast, the grid-based methods either minimize the entire landslide to a single representative point even for large

(>1 km$^2$) landslides or an array of points. Figures 3 and 4 show the final landslide maps of the Oregon watersheds and Puerto Rico, respectively, using the 30m sampling method for the grid-based maps and the slope unit-based maps using the median and SD predictor values with XGBoost. The other landslide maps are in Figures S5-S10. The slope unit maps generally better distinguish high and low probability zones with less area displaying probabilities near 0.5. Cumulative distribution functions of the maps' probabilities are shown in Figures S11 and S12.

Additionally, the slope-unit based maps are more granular, which prevents the more localized variation in probability present in the grid-based maps. This granularity generally results in a higher percent of study sites' areas displaying higher probabilities (Figure S13-S14). We note that the difference in map granularity is less for Puerto Rico than for the Oregon watersheds, likely due to the scale of mapped area, 30 m mapping unit, and the density of the landslide points (Figure 2). Finally, the different maps highlight similar locations within the watersheds as

having a relatively high or low probabilities.

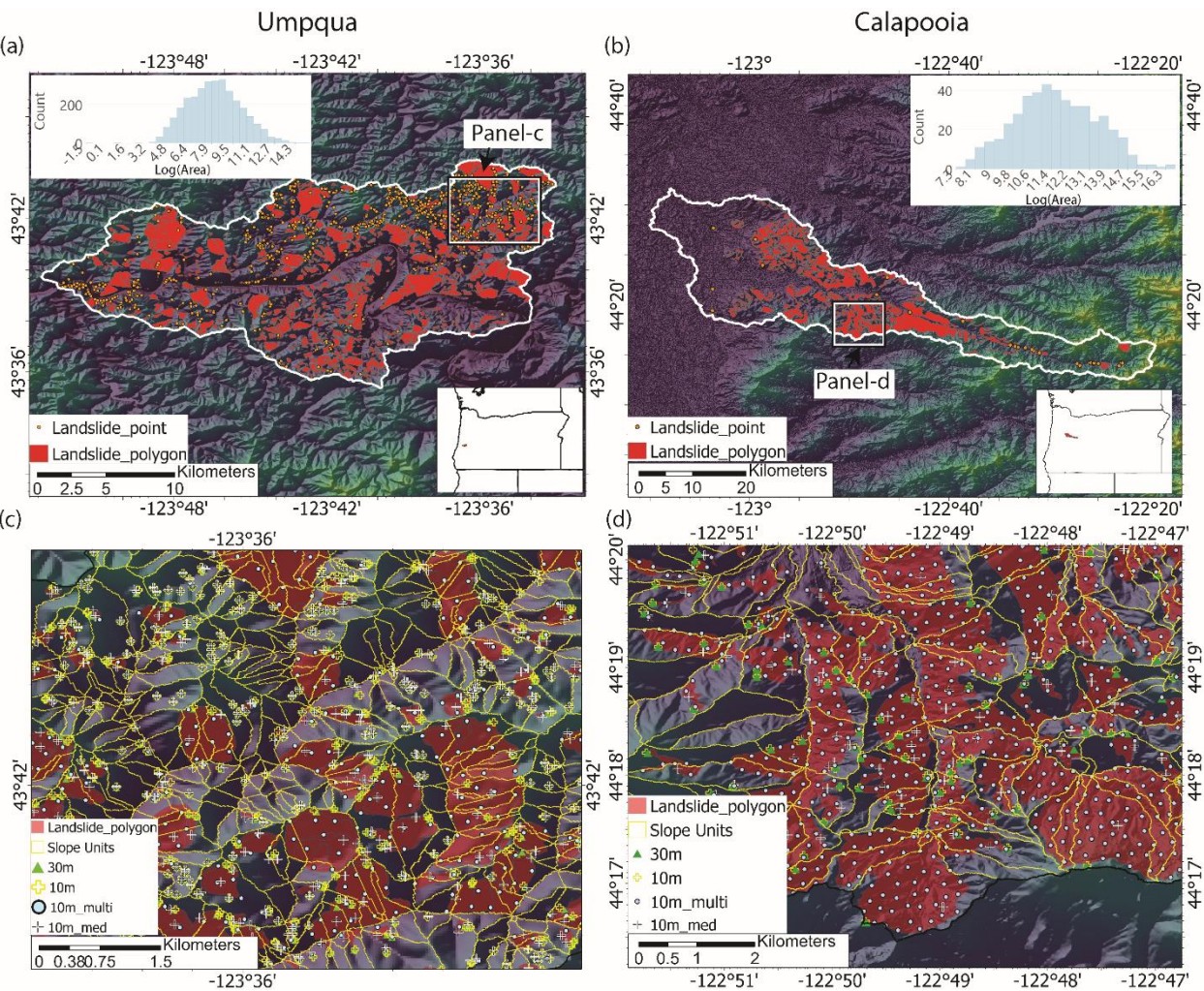

**Figure 1: Umpqua and Calapooia watersheds in Oregon. (a, b) digital elevation models and landslide inventories. Also shown are the log-normalized histograms of the landslide polygon areas. (c, d) zoomed-in portions of the slope unit maps with landslide polygons and grid sampled points using the four sampling**

 **techniques superimposed. The 10 m point samples often overlap the 30 m samples. Sampling techniques are described in section 2.2.**

**Figure 2: Island of Puerto Rico. (a) Slope unit delineation and mapped landslide points from Hurricane 335 Maria. (b) Zoomed--in portion of the island.**

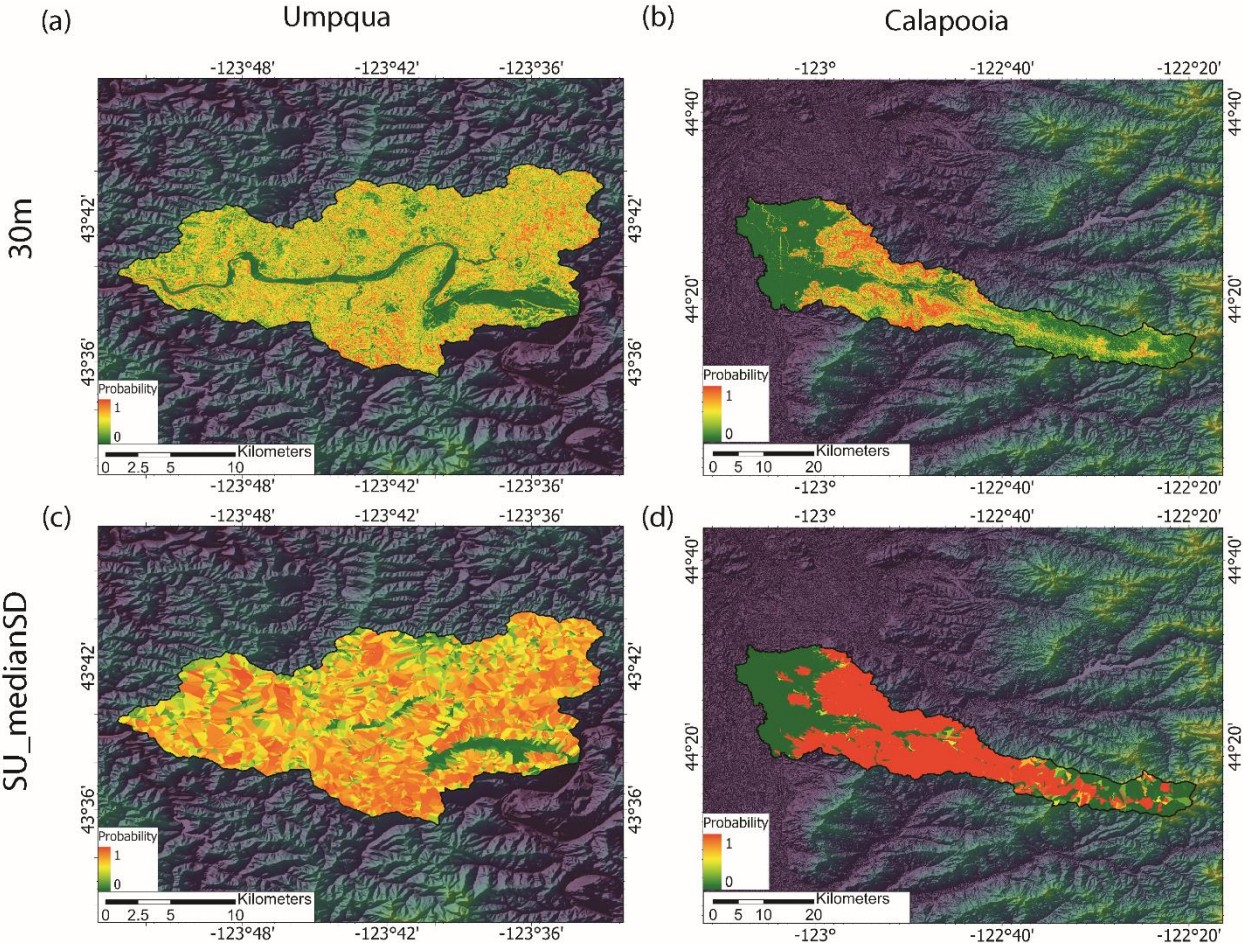

**Figure 3: Landslide susceptibility models for the Umpqua and Calapooia watersheds using (a,b) the 30m sampling method for the grid-based maps and (c,d) slope units with median and standard deviation predictor values (SU_medianSD) with XGBoost.**

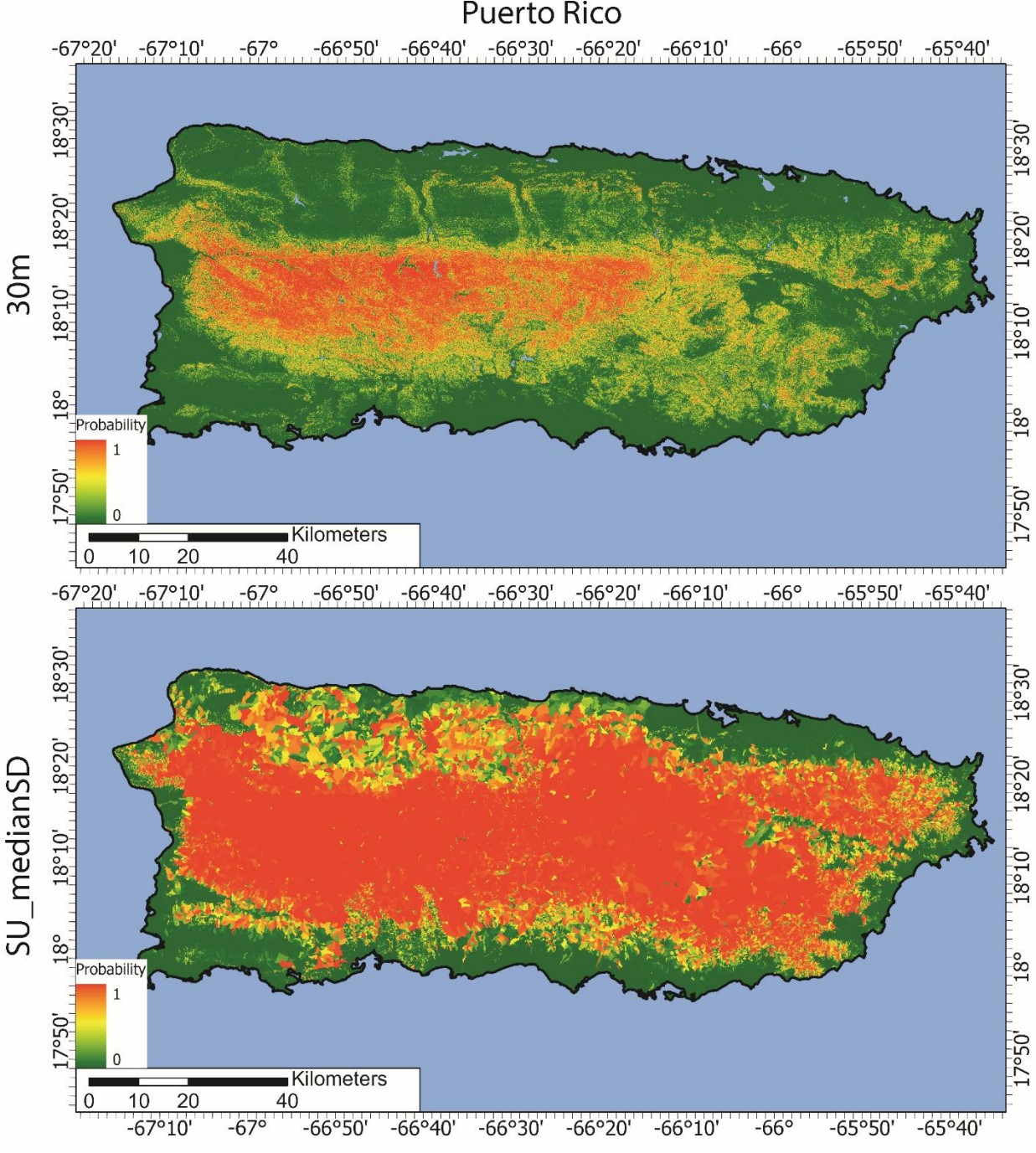


**Figure 4: Puerto Rico landslide occurrence models from the 30m grid-based maps and using slope units with median and standard deviation predictor values (SU_medianSD) with XGBoost.**

Both the ROC-AUC and Brier score metrics show a better model fit using slope units compared to any of the grid-based models for our study sites (Figures 5 and 6). The XGBoost and Logistic regression machine learning
algorithms show an increase in the median ROC-AUC and a decrease in the Brier scores for the slope unit-based maps. For example, at Calapooia, the XGBoost algorithm on the grid-based models showed AUC-ROC values that would qualify as very good model performance (average of 0.83) when applied to the test data, while the two final slope-unit based models had excellent performance (average of 0.96) when applied to the test data. The Brier scores of the same models applied to the test data demonstrate an average root-mean-square error of 0.17 and 0.07 for the

grid-based and slope unit models, respectively. Using the median and SD of the predictor values in each slope unit also increases the model performance compared to slope unit models developed with only the median predictor values. The different sampling techniques for the grid-based maps showed little variation in the two model performance metrics. Finally, XGBoost generally shows better model performance compared to logistic regression. In summary, the slope unit-based models can better differentiate high and low probability areas of the terrain.


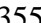

Figure 5: (a,b) Reciever operator characteristics (ROC)-area under the curve (AUC) and (c,d) Brier score boxplots from the 10-fold cross-validation procedure for landslide susceptibility models using the XGBoost (blue) and logistic regression (red) machine learning algorithms. The box hinges show the first and third quartiles; the whiskers extend to a maximum of 1.5 times the inter-quartile range; the red and blue dots show the data outlying the whiskers; and the horizonal bars show the median values of the distributions. Distributions are for the different sampling methods (10m, 30m, 10m_multi, 10m_med) and the slope unit


**(SU) maps using only the median (SU_medians) and the median and standard deviation of the predictor values (SU_medianSD). The black dots show the scores of the test datasets.**

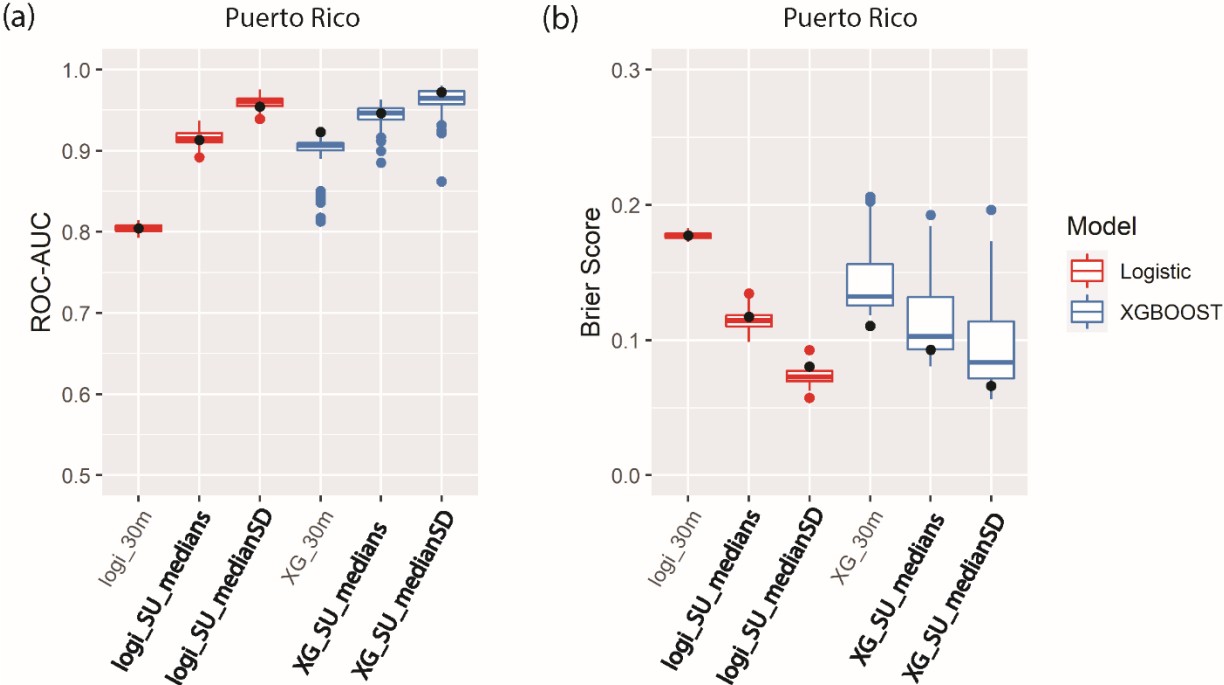

**Figure 6: (a) ROC-AUC and (b) Brier score boxplots from the 10-fold cross-validation procedure for landslide susceptibility models using the XGBoost (blue) and logistic regression (red) machine learning algorithms for the Hurricane Maria landslide catalog in Puerto Rico. Symbology is the same as Figure 5.**

**4 Discussion**

Our slope unit delineation algorithm, SUMak, has significant advantages over previous delineation methods. In contrast to other methods which use an optimization function or user-dictated setting for determining the appropriate scaling and positions of slope units, SUMak uses established geomorphic laws for determining an appropriate scale of the slope units to capture hillslope processes. This scaling provides a non-arbitrary scaling of the slope units that are optimized to capture hillslope processes and help prevent MAUP. Lastly, SUMak is computationally efficient compared to some other parameter-free algorithms. These advantages, coupled with it being open-source and easy-to-use, make it desirable for an array of geomorphic analyses.

Our analysis highlights some of the benefits and drawbacks of using grids or slope units for landslide susceptibility modeling when using landslide data with variable formats and no temporal component. While both methods generally highlight the same areas as being more susceptible, the 30 and 10 m resolution grid mapping units used in this study produce maps with smaller scale variations in susceptibility. While this level of detail can be advantageous, the vague nature of the susceptibility models' output caused by imprecise input data (e.g., no time component, imprecise locations, and variable formats) generally used to make susceptibility maps can cause misleading results. Indeed, producing high resolution (<100 m) grid-based maps is attempting to output results beyond the capacity of the input data. For example, in the Umpqua watershed, all the grid-based maps show only half of the terrain as having higher ($P > 0.5$) susceptibility (Figure S11). This phenomenon may partially reflect the limits of the statistical models used. However, slope units consistently produce more granular model results compared to grid-based maps independent of the model used, suggesting that the improved model performance is not merely an artifact of the statistical models. The lack of granularity of the grid-based maps at the Umpqua watershed may lead some to conclude that the watershed is generally not susceptible to landsliding. However, the abundance of the mapped landslides in the region (Figure 1b) indicate that most of the Umpqua watershed is highly

prone to landsliding. This shortcoming of the grid-based maps is also reflected in the poorer model metrics (Figure 5). In contrast, the larger mapping units available through slope units allows for a more conservative map that, we argue, better captures the level of susceptibility, even with imprecise input data. This is supported by the better model metrics (Figure 5) and a higher proportion of the Umpqua terrain as having higher susceptibility (Figures 3, S11, and S12). More conservative grid-based maps are generally achieved using larger grid cells, which accentuates the unrealistic geometry of the cells and exacerbates the imprecise mapping of susceptible areas. Thus, slope units provide an effective mapping unit that accurately delineates the terrain into slopes that can be used to create conservative susceptibility maps that better accommodate the nebulous output of regional susceptibility models created with inconsistent input data.

Slope units also provide a more conservative output for event-based landslide occurrence maps that may be more effective at communicating the likelihood of landsliding over large regions for some use cases. Like the maps created using non-temporal landslide datasets, the grid-based occurrence maps created for Puerto Rico show fine-scale variations in landslide probability that may be outputting results at too fine a resolution for the input data used to develop the model. This resolution results in high spatial heterogeneity of probability values within a single hillslope. Figure S15, shows a zoomed in portion of the model results and illustrates the diversity in probability values in the grid-based map compared to the slope unit map within a relatively small, mountainous terrain. The grid-based Puerto Rico landslide models are attempting to specify the pixel that contains the center of the head scarp. This level of precision may be useful for some purposes, but can be misleading and cause the model to miss the location of landslides induced by hurricane Maria. In contrast, the slope unit maps characterize the susceptibility of the entire hillslope and thus provide a more conservative output that better generalizes the location of hurricane-induced landslides. One tradeoff of using a larger mapping unit is that the model may assign the same high-probability value to the entirety of the slope unit even if landslides only affect a small portion of the slope unit. This can lead to maps that show larger areas as being more prone to landsliding compared to grid-based approaches; thus, slope units may not be appropriate for some landslide mitigation products.

Here we have focused on using slope units for statistical landslide susceptibility and near real-time landslide prediction modeling; however, objectively divided terrain can be used in an array of geomorphic studies. For instance, slope units could improve other landslide studies such as physically based models, early warning systems, debris flow modeling, or hazard assessments. These studies often use grid-based analysis which suffer from some of the same drawbacks of grid-based susceptibility modeling. Thus, adopting slope units as the mapping unit for these studies could yield more favorable results. Slope units could also help downscale topographically sensitive measurements (e.g., soil moisture, land cover, etc.) and provide a reasonable mapping unit for hydrologic and avalanche studies. Thus, SUMak could facilitate advances in geospatial analysis across several research areas beyond landslide susceptibility analysis.

**5 Conclusions**

The widespread use of slope units as the mapping unit of choice in landslide studies has been limited partially due to the lack of an efficient and easy-to-use method for delineating them. Here we introduce a new parameter-free algorithm for the automatic delineation of slope units. The algorithm is relatively computationally efficient and can be implemented anywhere there is digital elevation data. We also demonstrate that landslide maps created with slope units are more accurate and conservative compared to grid-based approaches.

**Code and data availability**

The code for SUMak and data used in this manuscript are available at Woodard (2023).

**Supplement link**

The supplement related to this article is available at: *future doi link*

**Author contribution**

JW developed the SUMak algorithm and drafted the paper. BM, NW, KA, BL, and MC reviewed the manuscript and contributed to the interpretation of the results.

**Competing interests**

The authors declare that they have no conflict of interest.

**Acknowledgments**

Any use of trade, firm, or product names is for descriptive purposes only and does not imply endorsement by the U.S. Government. We thank two anonymous reviews for their suggestions for improving the manuscript.

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
