# Peer review of "Slope Unit Maker (SUMak): An efficient and parameter-free algorithm for delineating slope units to improve landslide modeling"

_Natural Hazards and Earth System Sciences, 2023_

## Author Comment (AC1)

Below we have responded to each comment made by the reviewer. The reviewer's comments are in bold, and our responses are in roman text. Changes we have made to the original text are in italics.

**Review comment on the manuscript "Slope Unit Maker (SUMak): An efficient and parameter-free algorithm for delineating slope units to improve landslide susceptibility modeling" submitted to NHESS by Woodard et al.**

**GENERAL COMMENTS**

**Thank you for inviting me for this review. I have read the manuscript with great interest and I appreciate the effort to come up with a parameter-free tool for slope unit (SU) delineation.**

**In their manuscript, the authors present a newly developed tool for the automatic and parameter-free delineation of SU for landslide susceptibility mapping. In a first step, they compare SU created by their tool to SU created with the commonly used r.slopeunits algorithm by Alvioli et al., and in a second step, they compare the performance of landslide susceptibility models trained using different pixel-based and their SU-based landslide discretization methods for three case study sites to demonstrate the superiority of the SU-based approach as opposed to pixel-based methods.**

**I strongly believe in the advantages of SU as mapping units. I have used the r.slopeunits tool myself and found that the parametrization can be tricky and is not always transferable to other study areas. Thus, I believe that a parameter-free tool is an important step towards more objective and generalizable approaches in landslide susceptibility modelling.**

**As it comes to the comparison of SU vs. pixel-based approaches for landslide susceptibility modelling, it should be noted that there are already numerous publications on this topic.**

**Moreover, I am not totally convinced by the discussion of the SU result presented in Fig. 1 and by the attempt to demonstrate the superiority of the SU-based approach. This is partly because there is some information lacking that would enable the readers to clearly interpret the results. But also, the discussion appears a bit brief and shallow, considering the authors compared so many different approaches in three study areas. I would expect a deeper and more critical discussion not only focusing on performance metrics, but also model set-up and spatial performance for all case studies, also the Puerto Rico one. Please find some specific comments and questions below.**

We thank the reviewer for their careful reading of our manuscript and are pleased that they appreciate the importance of creating a parameter-free slope unit delineation tool. We have removed our direct comparison to r.slopeunits to better focus the manuscript and to avoid any unfair comparisons between the two delineation tools. We have added more details to the manuscript to better describe our model setup and the datasets used in our analysis. Please see our responses below to your comments on section 2 for more details. A few additional adjustments to better discuss our method and results are described here.

We have revised the introduction to discuss the importance of an appropriate scaling of slope units. It reads as follows:

*The scaling of slope units should not be arbitrarily set to avoid the modifiable areal unit problem (MAUP) (Openshaw and Taylor, 1983; Buzzelli, 2020; Goodchild, 2011). The MAUP occurs when the cartographic representation of data varies significantly by the scale of the mapping unit used to represent the data. MAUP is a challenging issue to overcome; however, determining a scale of the slope units so that they effectively capture the hillslope processes of interest can greatly mitigate the negative effects of the MAUP (Buzzelli, 2020). Alvioli et al. (2020) recognized this challenge, which motivated the development of their custom optimization procedure. Importantly, the optimal scale for capturing hillslope processes is spatially variant. Thus, the ideal scaling of slope units should adjust to the local topography.*

We add to this point at the end of section 2.1 where we write the following:

*We argue that basing the scaling of slope units used for landslide analysis on established geomorphic laws provides the best justification for their appropriate sizing and odds of mitigating the negative effects of the MAUP.* Further details on how the algorithm was implemented in R are in Text S1 *and the online repository (Woodard, 2023).*

We follow up this discussion in the first paragraph of section 4. It reads as follows:

Our slope unit delineation algorithm, SUMak, has significant advantages over previous delineation methods. In contrast to other methods which use an optimization function or user-dictated setting for determining the appropriate scaling and positions of slope units, SUMak uses established geomorphic laws for determining an appropriate scale of the slope units to capture hillslope processes. *This scaling provides a non-arbitrary scaling of the slope units that are optimized to capture hillslope processes and help prevent MAUP.* Lastly, SUMak is computationally efficient compared to *some* other parameter-free algorithms. These advantages, coupled with it being open-source and easy-to-use, make it desirable for an array of geomorphic analyses.

To better describe the details on the model set-up and spatial performance we augmented section 2.2 in several locations. The first paragraph now reads as follows:

[revised manuscript text omitted]

We also added a figure to better illustrate the slope unit binary classification of landslide existence (see our response to Line 200 below).

**Also, some maps are in my opinion not ideally composed.**

We have removed figure 1 from the manuscript to better focus our analysis. We have remade figure 2 (now figure 1) per the reviewer's recommendations below and moved the slope units of the entire watersheds to larger figures in the supplemental information document (Figure S4). We also adjusted the color of the slope units in Figure 3 to improve their visualization.

[Figure]

(a) Umpqua

Elevation (m)
1297
0
Slope Units
Kilometers
0  2.5  5  10

(b) Calapooia

Elevation (m)
1811
-18
Slope Units
Kilometers
0  5  10  20

**Figure S4.** SUMak delineated slope units over the (a) Umpqua and (b) Calapooia watersheds, Oregon.

[Figure]

**Figure 2:** Island of Puerto Rico. (a) Slope unit delineation and mapped landslide points from Hurricane Maria. (b) Zoomed--in portion of the island.

**Apart from that, the English language is flawless, the paper is well structured, and the references are complete. Multiple references in the text should be sorted alphabetically though.**

We appreciate the positive feedback. We have double checked the sorting of references.

**I think that after the additional information on the methodology has been provided and the discussion improved, the paper could be published.**

**SPECIFIC COMMENTS**

**Abstract**

**I would suggest to mention the software the proposed algorithm runs on in the abstract (and also in the main text in section 2.1). This is relevant information for the readers.**

We rewrote part of the abstract to specify the software used by the algorithm. It now reads as follows:

We introduce a computationally efficient algorithm for the parameter-free delineation of slope units *that leverages tools from within TauDEM and GRASS, using an R interface.*

In section 2.1 we specify how the SUMak algorithm leverages different software packages.

**1 Introduction**

**Lines 120-127: Could you mention here which methods were used for landslide susceptibility modelling?**

We inserted the following sentence into this section.

*Landslide models were developed using logistic regression and XGBoost machine learning algorithms.*

**2 Methods**

**Lines 130-143: Since the new method is presented as "easy-to-use" I would expect a little more information and instructions on where and how to run it for readers who are not so familiar with GRASS or R for geospatial analyses, or at least a reference to the repository where more detailed instructions can be found.**

We amended this paragraph to include a reference to the online repository that details how to run the software within R. The last sentence of this paragraph now reads as follows:

Further details on how the algorithm was implemented in R are in Text S1 *and the online repository (Woodard, 2023).*

**Lines 144-147: Which parameters were used for the r.slopeunits algorithm?**

Per the recommendation of the other reviewer, we have omitted this direct comparison with r.slopeunits.

**Lines 163-169: What types of landslides were included in the invetories?**

We included the following sentence to address this point:

*Each area's landslide catalog includes an assortment of landslide types (slumps, debris flows, rockfalls, deep-seated landslides, and others) which are not differentiated in this study.*

**Lines 167-168, lines 179-180: It is a bit unclear to me. What I understand is that the landslide inventories were mixed, with some landslides represented as points, and others as polygons, and the points were mapped at the centroids of the landslides. How many points and polygons, respectively, did each of the landslide inventories contain? How did you deal with landslides that were originally mapped as (centroid?) points for the different sampling strategies that put the points at the scarp or randomly within a landslide body?**

We added the number of landslide points and polygons to this paragraph. It now reads, in part, as follows:

*The Umpqua dataset contains 941 points and 3213 polygons, while the Calapooia dataset contains 33 points and 456 polygons.*

Further down the paragraph, it now reads,

The Puerto Rico landslide dataset consists of *71,431* point locations of the centers of landslide headscarps that occurred during Hurricane Maria on September 20-21, 2017 (Hughes et al., 2019).

The median point location conversion method allows the landslide data to be more compatible with the centroid mapping method used in the Oregon datasets. However, as only a minority of landslides in the different datasets were mapped as points (29% and 7% for Umpqua and Calapooia, respectively), we did not pursue an additional standardization method using the centroids of all the landslide polygons.

**Lines 200-201 and lines 202-209: It would be very helpful for the interpretation of the modelling results if you could provide some statistics. How many samples did each dataset contain? How many SU were delineated in each study area? What was the original positive to negative ratio, especially for the SU?**

We have added the number of landslides of each datatype to section 2.2. Please see our response to your previous comment for details. We have included the number of slope units to Table 1. Finally, we added additional figures to the supplemental (Figures S2 and S3) that shows which slope units do or do not contain a landslide, per your recommendation in your final comment below. This illustrates the positive to negative ratio and facilitates interpretation of the model results.

**Table 1.** SUMak performance metrics.

| Location | Area (km²) | Coastline | DEM Resolution (m) | Compute Time (minutes) | Slope Unit Count | Time per area (seconds/km²) | Time per Slope unit (seconds) |
|---|---|---|---|---|---|---|---|
| Umpqua | 257 | No | 10 | 3.11 | 3841 | 0.7 | 0.05 |
| Calapooia | 743 | No | 10 | 9.97 | 6990 | 0.8 | 0.09 |
| Puerto Rico | 8870 | Yes | 10 | 383.28 | 140367 | 2.6 | 0.16 |

**Umpqua**

[Figure]

**Calapooia**

Figure S2: Maps illustrating the existence (red) or non-existence (green) of a landslide within each slope unit over the Umpqua and Calapooia watersheds, Oregon.

[Figure]

Figure S3: Maps illustrating the existence (red) or non-existence (green) of a landslide within each slope unit over Puerto Rico.

**Line 210-229: Which software was used for the susceptibility modelling? Did you conduct any data preparation, such as scaling? Why didn't you use lithology as an input parameter?**

We inserted the following at the end of section 2.2:

*Model development and post-processing was conducted within R (R Core Team, 2016).*

We did not perform any data preparation beyond what we describe in the text.

We did not use lithology as an input parameter because we did not have a consistent lithological map of all the areas that has sufficient resolution to be useful.

**Section 2.2: How were the final landslide susceptibility maps generated? Were the trained models applied to all pixels in the study area in the pixel-based approaches? And for the SU-based approach, did you apply the trained model on a pixel-basis or SU-basis?**

We applied the trained models to entire study areas to create the final maps. Models were applied to all the pixels or slope units depending on the mapping unit used to train the model. To clarify this point we added the following to the end of the 4th paragraph of section 2.2.

*After the models are trained we generated maps by applying the trained models to the entire study areas.*

**3 Results**

**Fig. 1: The different scales of the two excerpts are confusing. What are the colors in map c? In case they are SU, its unrecognizable. A plain hillshade or DEM could work better.**

We have removed this figure per the recommendation of the other reviewer.

**Lines 259-260 and Fig. 1: To me the SUMak SU look much more heterogenous than the r.slopeunits ones. Some SU are larger, and then there are some areas containing many small ones. Could you explain this in more detail? Is the result really so similar to the r.slopeunits one? Here it would also help to know which parameters were used for the latter, see my previous comment.**

We have removed this figure and text per the recommendation of the other reviewer.

**Fig. 2 a, b, Fig. 3 a: at these scales it is impossible to recognize the SU. I would suggest to enlarge the maps or omit them. Then again, for being able to interpret the performance of the landslide susceptibility maps, it would be helpful to see maps with the distribution of positive and negative SU.**

We have removed fig. 2a,b, instead putting a larger figure of the slope units of the entire watersheds in the supplemental (Figure S4). We changed the colors of the slope units in Figure 3. While the slope units are still difficult to see in 3a, it provides a reference map for figure 3b and provides a figure illustrating the distribution of landslide points associated with Hurricane Maria. We provide zoomed in portions of the slope unit maps to make the slope unit more recognizable (2c,d and 3b). We have also created an additional figure that shows a binary categorization of the slope units for comparison to the final landslide maps (Figures S2 and S3). See these new figures above.

---

## Author Comment (AC2)

Below we have responded to each comment made by the reviewer. The reviewer's comments are in bold, and our responses are in roman text. Changes we have made to the original text are in italics.

**The manuscript nhess-2023-70 proposed for publication in NHESS describes a new algorithm for slope unit delineation. Slope units are a well-known terrain subdivision type in the landslide research community; the topic of the manuscript is well within the aims & scope of the Journal. The authors make a good case about the use of slope units in landslide susceptibility mapping. They describe the advantages of using slope units in conjunction with statistical methods, discussing the use of heterogeneous data, landslide inventories of varying quality, data inaccuracies, and drawbacks of using grid cells.**

We thank the reviewer for their thorough and thoughtful comments and are pleased that they appreciate the importance of our manuscript.

**The paper is intentionally split into a rather technical part, about the outcomes of the software introduced here, and an application part, about use of slope units for landslide susceptibility mapping. I have two main general comments about those parts, and a few specific comments that I will list afterwards.**

**The main issue in the technical part in my opinion is that it is rather focused on the comparison of the outcome and, above all, speed of the new software with respect to the existing r.slopeunits software by Alvioli et al. I believe the way this part is presented is a bit misleading, because it makes assumptions and comparisons that may be not entirely justified. My understanding is that the main difference between the two pieces of work are that the previous one require parametric inputs, while the one presented here only gives one possible result, with no additional parameters. (The new software is not described in detail, so it is difficult to be more specific on the input requirements). Because of this difference the authors stress in several occasions that the existing software has much larger computational demands, and it has "prohibitive" processing times. In an explicit comparison of the outputs of the two software in Sicily (a part of Italy where the slope unit map were published by the r.slopeunit team) the authors of the new software estimated the processing and optimization time based on the total running time quoted in the original paper for the entire Italy, scaling it down proportionally to the size of the area. I felt that as an unfair comparison and, being myself a user of r.slopeunits, it was easy to run the software a couple of times with different values of the input parameters (on the same 25 m EU-DEM). The two runs required 100 minutes and 140 minutes with typical values of parameters I usually input, using a maximum of about 2GB of RAM, on a single computing core (a rather outdated CPU, to tell the truth). To compare with the computing time quoted by the authors of the new software (7 hours), who used 16 computing cores, running about 48 instances of r.slopeunits if would require three times that; say, an average of about 120x3 = 360 minutes. Thus in four hours one could have done about 48 runs, and picked up the "best" one, within the criteria developed by the r.slopeunit team. I would say that the difference is mostly due to the fact that the quoted processing time in the previous paper was due to a rather complicated optimization algorithm, taking into account a huge number of runs and some peculiar arrangement of optimization spatial domains. Then I**

**would go on and say that this is not a fair comparison either, because the outputs of the two pieces of software seem very different, judging from figure 1 in the manuscript: one can clearly see that the slope units obtained with the new software are different from the existing ones, and to be honest I can hardly say that they provide a comparable accuracy in segmenting the sub-areas: one can see areas with very similar morphological setting that are split into tiny details by SUMak, where the previous delineation looks more "reasonable". There probably are areas with the opposite situation, even if they are more difficult to spot. So, different output, different computing times and rather different flexibility - the possibility of having different outputs with different input requirements looks like an advantage, in my opinion, because it gives the possibility of tuning the map to one's needs. In conclusion, and in essence, my suggestion is either to substantially improve the comparison part (I will not suggest to try and compare with other methods - even if I believe other methods exist that were defined as parameter-free by their respective authors), or to reduce its relative importance in the manuscript, in favour of the practical application part.**

The reviewer is correct that the use of r.slopeunits is a valid approach in circumstances where the user is not concerned about obtaining the optimum sizing of the slope units in the study area or when a custom optimization function is desired. However, slope units are often delineated and chosen at an arbitrary scale that reduces reproducibility and complicates the utility of slope units as a mapping unit for landslide susceptibility analysis. The reviewer is also correct that it is the optimization procedure and cleaning algorithm used by Avlioli et al. (2020) that drastically increased the computational time for delineating slope units over Italy. However, as argued by Alvioli et al. (2020) this optimization procedure is critical. They state, "Identifying the correct scale of a particular spatial analysis is a way out from what is known as modifiable areal unit problem (Openshaw (1984); Manley (2014)). Any study associated with the use of data aggregated within geographical areas is prone to the MAUP, and an objective link between mapping units and the underlying topography is highly desirable."

To address the reviewer's point, we have augmented the introduction to better highlight the importance of the proper scaling of slope units. The second to last paragraph of the introduction now reads as follows:

*The correct scaling (size) of slope units should not be arbitrarily set to avoid the modifiable areal unit problem (MAUP) (Openshaw and Taylor, 1983; Buzzelli, 2020; Goodchild, 2011). The MAUP occurs when the cartographic representation of data varies significantly by the scale of the mapping unit used to represent the data. MAUP is a challenging issue to overcome. However, determining a scale of the slope units so that they effectively capture the hillslope processes of interest can greatly mitigate the negative effects of the MAUP (Buzzelli, 2020). Alvioli et al. (2020) recognized this challenge, which motivated the development of their custom optimization procedure. Importantly, the optimal scale for capturing hillslope processes is spatially variant. Thus, the ideal scaling of slope units should adjust to the local topography.*

We also agree that the comparison to r.slopeunits is ancillary to the main objectives of the paper. We have removed this portion of the methods and results in favor of a paragraph in the results

that describes the speed and other specifications of our algorithm without the in-depth comparison to r.slopeunits. Section 3.1 now reads as follows:

**3.1** *SUMak Slope Unit Delineation*

*SUMak is able to quickly delineate slope units over the three study areas while automatically adapting the scaling of the slope units by the local terrain. Table 1 shows the time to delineate each of the study areas. Both Oregon watersheds were delineated in only a few minutes while the island of Puerto Rico took substantially longer. This is due to the larger area and the increased complexity of the delineating watersheds near coastlines where watersheds get increasingly small due to decreased accumulation areas. The adaptation of the slope unit sizes to the local topography is apparent in the slope unit maps (Figures S4, 1-2). For example, the Calapooia Watershed includes a mountainous and flat region (Figure 1). SUMak created smaller slope units over the flat region compared to the mountainous region to accommodate the difference in scale where hillslope processes occur (Figure S4).*

| *Table 1.* SUMak performance metrics. | | | | | | | |
|---|---|---|---|---|---|---|---|
| *Location* | *Area (km²)* | *Coastline* | *DEM Resolution (m)* | *Compute Time (minutes)* | *Slope Unit Count* | *Time per area (seconds/km²)* | *Time per Slope unit (seconds)* |
| *Umpqua* | *257* | *No* | *10* | *3.11* | *3841* | *0.7* | *0.05* |
| *Calapooia* | *743* | *No* | *10* | *9.97* | *6990* | *0.8* | *0.09* |
| *Puerto Rico* | *8870* | *Yes* | *10* | *383.28* | *140367* | *2.6* | *0.16* |

**The authors proposed two applications of the slope unit maps produced with the new software, in two different areas. The two applications are substantially different: in the one case (actually two, in Oregon, USA), they authors use landslide data collected over decades, and the the other case (island of Puerto Rico) landslides were all caused by a hurricane - thus, a specific event. I believe that many would question the use of an event-based landslide inventory to map landslide susceptibility, without mentioning the necessary caveats. In fact - if we agree that landslide susceptibility is the spatial component of landslide hazard - it is difficult to fit an event-based map within this definition. While a "generic", or "historical" landslide inventory actually tells something about the different likelihood of landslides to occur at different locations in a study area, an event-based landslide inventory clearly does a very different job. This is very clear from the maps obtained in the two areas, with statistical methods; but this is in contradiction with the statement "statistical models analyze the spatial distribution of known landslides in relation to local terrain condition", because there no terrain condition alone can explain the landslide distribution in figure 3, and the susceptibility map in figure 5. The authors stated all of this in one line (228), just mentioning that they included soil moisture data as an additional predictor. This singles out the map obtained from the event data as something different from a susceptibility map. For example, in a few papers, this was done**

**using shake maps to account for a trigger in co-seismic landslides. There, the ground shaking parameters are interpreted somewhat as a dynamical input; examples are in Nowicki et al. (10.1016/j.enggeo.2014.02.002) and in Tanyas et al. (10.1016/j.geomorph.2018.10.022). In conclusion, I believe this aspect has been overlooked by the authors of the proposed manuscript, and it should be discussed in some detail as the different interpretation and purpose of the results is not obvious. Another, related point is the repeated reference to the absence of a time component in input data; how would it change the outcome? That would require a totally different framework, in my opinion.**

We have rewritten several portions of the text to both clarify and elaborate on the differences between Oregon and Puerto Rico examples. We included a new introductory paragraph that explains the difference between the two datasets. It reads as follows:

*Another tool used to mitigate losses associated with landslides are near real-time or forecasted assessment of event-specific landslide occurrence models (Nowicki Jessee et al., 2018; Nowicki et al., 2014; Tanyas et al., 2019; Kirschbaum and Stanley, 2018). Rather than characterizing the potential of landslide existence from static terrain conditions, these models include a dynamic input designed to characterize landslide potential from a particular forcing event. For example, Tanyas et al. (2019), analyzed the static terrain conditions and dynamic ground motion metrics (e.g., peak ground velocity) from 25 earthquake-induced landslide-event inventories from across the world to create a landslide model that can estimate the distribution of landslides during an earthquake. Herein, we will refer to this model type as landslide occurrence models. Like susceptibility models, landslide occurrence models often suffer from imperfect and heterogeneous landslide data. Thus, a common problem in the landslide community is determining an effective way of assessing landslide susceptibility and/or occurrence, despite the imperfect data available for model development.*

The first paragraph of section 2.2 now reads, in part, as follows:

[revised manuscript text omitted]

Finally, we have made several minor changes to the text to avoid confusion of the uses of the two maps.

The references to input data often missing a time component was intended to exemplify the vague character of most landslide catalogs used to create landslide susceptibility maps. We believe that our previous adjustments to the text have resolved the reviewer's last point in this general comment.

**Minor or not-so-minor comments follow.**

**As anticipated, the abstract is very unbalanced towards the computational speed of the codes, rather than on actual methods and conclusions of the proposed work. As similar comments apply in different parts of the text, I will not point to all of them - only I would like to add that one fair comment can be that regardless of the few or many hours needed to prepare a slope unit map, that is a one-time effort and processing time may be less important than other aspects.**

We have rewritten the abstract to deemphasize the computational speed comparison with other methods. We did the same for the short summary and elsewhere in the text. The abstract and short summary now read as follows.

**Abstract.**
Slope units are terrain partitions bounded by drainage and divide lines. *In* landslide modeling, *including susceptibility modeling and event-specific modeling of landslide occurrence, slope units provide several advantages over gridded units*, such as better capturing terrain geometry, improved incorporation of geospatial landslide-occurrence data in different formats (e.g., point and polygon), and better accommodating the varying data accuracy and precision in landslide inventories. However, the use of slope units in regional ($>$100 km$^2$) landslide studies remains limited due, in part, to *the large* computational costs and/or poor reproducibility with current delineation methods. We introduce a computationally efficient algorithm for the parameter-free delineation of slope units *that leverages tools from within TauDEM and GRASS, using an R interface.* The algorithm uses geomorphic scaling laws to define the appropriate scaling of the slope units representative of hillslope processes, avoiding the *often ambiguous determination of slope unit scaling.* We then demonstrate how slope units enable more robust regional-scale landslide susceptibility *and event-specific landslide occurrence* maps.

**Short summary**
Dividing landscapes into representative hillslopes greatly improves predictions of landslide potential across landscapes *but their scaling is often arbitrarily set and can require significant computing power to delineate.* Here, we present a new computer program that can efficiently divide landscapes into meaningful slope units *scaled to best capture landslide processes.* The results of this work will allow an improved understanding of landslide potential across different landscapes and can ultimately help reduce the impacts of landslides worldwide.

Please see our response to your first general comment for more examples of how we have adjusted the text to comply with this suggestion.

**In section 2.1 the authors refer to the "constant drop law"; I feel it would be nice to have a bit more on this law and its meaning. In addition to a quantitative description it would be nice to explain why is the law relevant for slope unit delineation. Do hillslope processes uniquely determine slope unit boundaries, or are they more relevant to identify landslides? Are slope units produced by this criterion suited for any kind of landslides, or for a subset of them? Did the authors check that the law actually holds, in the specific areas investigated here and - more importantly - with the representation of the terrain provided by the digital elevation models adopted here? Moreover, another relevant point is - is this criterion applicable to areas of any size? This is probably the motivation behind the different optimization procedure adopted by Alvioli at al 2020 (and further refined in Alvioli et al 2021 - 10.1080/17445647.2022.2052768).**

We have rewritten section 2.1 to address the concerns about the reliability of TauDEM and the appropriateness of our selected scaling for characterizing landslide hazards. It reads as follows:

This scaling law is independent of the raster resolution (Tarboton et al., 1991; Tarboton, 1989) *and has been used extensively in the field of fluvial geomorphology.* We further process these optimally scaled watersheds by splitting them by the longest flow path within the watershed using GRASS (GRASS Development Team, 2020). Thus, the watersheds essentially become what would be objectively recognized as a slope. *We argue that basing the scaling of slope units used for landslide analysis on established geomorphic laws provides the best justification for their appropriate sizing and odds of mitigating the negative effects of the MAUP.* Further details on how the algorithm was implemented in R are in Text *S1 and the online repository (Woodard, 2023).*

Based on the extensive validation procedures conducted by the Tarboton (1989) and Tarboton et al. (1991) and the sound theoretical basis of the drop law, we do not perform further validations for our areas of interest.

The criterion is applicable to areas of any size as it will find the desired threshold for flow accumulation across area of interest. However, the size of the area of interest can alter the calculated threshold if it includes a significant variation in topographic signatures (e.g.., mountains and alluvial fans. We discuss this in the second paragraph of the supplemental text 1 where we state the following:

"Creating intermediate watersheds allows the algorithm to adapt the scaling of the slope units according to the characteristics of the local topography. If the intermediate watershed has significant variation in topography, TauDEM may choose a threshold that doesn't adequately characterize every area within the watershed. Thus, intermediate watersheds must be small enough to limit the variation in topography but large enough to avoid significantly reducing computational efficiency. While experimenting with different watershed dimensions on the topographically diverse regions of Sicily, Puerto Rico, and the Umpqua and Calapooia

watersheds, we found an accumulation threshold of ~100 km$^2$ to adequately strike this balance. This threshold can be adjusted to meet the user's needs, or SUMak has an option to input predetermined intermediate watersheds. After appropriate intermediate watersheds are created, the algorithm runs the rest of the processing steps individually for each intermediate watershed in parallel. "

**About the conversion of landslide polygons to point-like features: it is not very clear to me why this is necessary or, actually, what is the rationale of the different methods, especially converting one polygon into multiple points; I can understand using the highest elevation point as an indicator for the landslide initiation point - but what is the difference in using equally-spaced multiple points instead of all of the grid cells overlapping a landslides? Maybe I am missing something, here.**

We have revised this section to better explain our rational for using multiple points per polygon. It now reads as follows:

Creating multiple points within the polygons allows us to capture some of the variability in the large landslides' measured attributes *without eliminating the influence of landslides originally mapped as points. Using all the raster cells within the polygons would essentially oversaturate the model with data from the landslide polygons and omit any influence of the landslides originally mapped as points.*

**When introducing the XGBoost method - what is the meaning of the list of parameters (max_depth) and most importantly how does the optimization work, in short?**

We have added a short description of the XGBoost optimization procedure. It reads as follows:

To increase the model accuracy while preventing overfitting, we optimize the 'max_depth', 'min_child_weight', 'subsample', 'gamma', and 'colsample_bytree' hyperparameters of XGBoost (see Chen & Guestrin, 2016 and https://xgboost.readthedocs.io/ for an explanation of these parameters) using a Bayesian cross-validation procedure. *In short, these hyperparameters adjust how the model adapts to fit the training data. The Bayesian cross-validation procedure uses ten folds and ten iterations and uses the results from the previous iterations to inform the next iteration of hyperparameters to use (Snoek et al., 2012). This procedure prevents the use of unwieldly grid searches and permits faster optimization of the model hyperparameters.*

**I do not understand the sentence "the Brier score provides measure of the scale of the model fit and not just its ordering"; can the authors explain, in short?**

We rewrote this sentence to clarify our meaning. It now reads as follows:

Thus, a *B* value of zero suggests perfect model fit and a value of one indicates perfect misfit. In contrast to AUC-ROC, the Brier score provides measure of the scale of the model fit and not just its ordering of *landslide and non-landslide observations.*

**As already mentioned, the method to scale down the computing time in lines 265-267 does not seem reasonable at all, for a quantitative nor qualitative comparison.**

We have removed this portion of the text.

**I believe the overall view of slope units in figure 1(c) is rather poor - most probably due to the attempt of showing the slope unit vector layer at that zoom scale. On similar grounds, it is kind of impossible to see anything sensible in figure 2(a)-(b). Slope units would be much more visible in figure 3(b) if they weren't colorized as the higher elevation values.**

We have removed figure 1 from the text to better focus the scope of the manuscript. We have also removed figure 2a,b and changed the colors of the slope units in figure 3 to facilitate visualization. The updated figures are copied below.

[Figure]

**Figure 2:** Umpqua and Calapooia watersheds in Oregon. (a, b) digital elevation models and landslide inventories. Also shown are the log-normalized histograms of the landslide polygon areas. (d, d) zoomed-in portions of the slope unit maps with landslide polygons and grid sampled

points using the four sampling techniques superimposed. The 10 m point samples often overlap the 30 m samples. Sampling techniques are described in section 2.2.

[Figure]

**Figure 3:** Island of Puerto Rico. (a) Slope unit delineation and mapped landslide points from Hurricane Maria. (b) Zoomed--in portion of the island.

**The discussion about the percentage of grid cells with large susceptibility values, which result in visually under-represented with respect to the number and size of landslides, is interesting. The authors ascribe that to a poor performance of methods based on grid cells, and better suitability of the slope unit approach. Could it be that the statistical methods themselves reveal their limits? The observed overall picture is seemingly typical of over-fitting, for methods with poor generalization performance.**

Yes, that is one reason why we used two different algorithms in our analysis. However, the ability of slope unit-based maps to better differentiate high and low susceptibility zones remains. To address this point, we inserted the following into the second paragraph of section 4:

*This phenomenon may partially reflect the limits of the statistical models used. However, slope units consistently produced more granular model results compared to grid-based maps independent of the model used, suggesting that this is not merely an artifact of the statistical models. The lack of granularity of the grid-based maps at the Umpqua watershed may lead some to conclude that the watershed is generally not susceptible to landsliding.*

**Despite the interesting premises set by the introduction and discussion sections, the conclusions drawn by the authors do not seem to meet the expectations (at least, my expectations). I mean, the difference between slope units vs. pixel based models has been investigated by several authors, with similar conclusions. Maybe a bit more could have been done in highlighting the role of the optimization algorithm, for the slope unit delineation part, and on the meaning of using a landslide inventory corresponding to an individual event instead of a "generic" inventory, for the susceptibility part.**

We believe that our adjustments made in response to the second minor comment and the second general point address this concern.

---

## Author Response (AR2)

Below we have responded to each comment made by the two reviewers. The reviewers' comments are in bold, and our responses are in roman text. Changes we have made to the original text are in italics.

Reviewer #1 Comments
* * *
**Review comment on the manuscript "Slope Unit Maker (SUMak): An efficient and parameter-free algorithm for delineating slope units to improve landslide susceptibility modeling" submitted to NHESS by Woodard et al. – round 2**

**GENERAL COMMENTS**
**Thank you for inviting me to reevaluate this manuscript in the second round of reviews. I checked the replies of the authors and the modified manuscript carefully. I want to thank the authors for providing more explanations and improving the manuscript and figures according to my suggestions. I agree with most of the changes, with a few follow-up questions below.**

We thank the reviewer for their initial review and their careful follow up.

**My main point of critique is still the discussion, which is only based on comparing statistical performance measures and the granularity of the resulting maps (in my opinion a no-brainer when comparing pixel-based to SU-based models). It has not been improved much in the revised version. There is no connection between the statistics and the geological phenomenon. In which areas did the false positives and false negatives occur and what might be geological explanations for it? Which role does the original landslide distribution that is varying significantly between the three study areas, play? These and many other questions could be addressed, in order to generate new understanding of spatial patterns of landslides. Anyway, as it is, the discussion does serve the purpose of demonstrating the superiority of the SU-based models, so maybe I am asking too much.**

The reviewer poses many interesting questions that could provide many insights into the landslide controls at our three study sites. However, as the reviewer notes, these questions are ancillary to the main objectives of our manuscript, which are to introduce and demonstrate the utility of SUMak for delineating slope units. Consequently, we do not address these questions at this time.

**Please check for some grammar artifacts that were now added to the text.**

We have reread through the manuscript several times to correct the grammatical errors in the revised manuscript. Please see these changes in the tracked-changes document for edits.

**SPECIFIC COMMENTS**
**Original comment:**
**Lines 167-168, lines 179-180: It is a bit unclear to me. What I understand is that the**

**landslide inventories were mixed, with some landslides represented as points, and others as polygons, and the points were mapped at the centroids of the landslides. How many points and polygons, respectively, did each of the landslide inventories contain? How did you deal with landslides that were originally mapped as (centroid?) points for the different sampling strategies that put the points at the scarp or randomly within a landslide body?**

**Follow-up question: I still do not understand what happened to the landslides that were only in the database as centroid points. Were they omitted in the different pixel-based sampling approaches? Or were they omitted in all approaches? I just wanted to make sure all models were based on the same distribution of landslide cases.**

At the beginning of the second paragraph of section 2.2 we state the following:

"We evaluate four different methods of standardizing landslide polygons to points for grid-based susceptibility maps in the Oregon watersheds. Each method converts the polygons to points *which* are combined with the landslides originally mapped as points."

There was a typo at the beginning of this paragraph which may have led to some confusion. However, to reiterate, the landslides initially mapped as points were retained for all the different sampling approaches. Landslide polygons converted to points were thereafter consistent with landslides mapped as points, and merged with the original, points-based dataset.

**Original comment:**
**Lines 200-201 and lines 202-209: It would be very helpful for the interpretation of the modelling results if you could provide some statistics. How many samples did each dataset contain? How many SU were delineated in each study area? What was the original positive to negative ratio, especially for the SU?**

**Follow-up question: Could you please also add the percentages of the SU classified as positive and negative, respectively and the number of training samples each model was based on? I understand that they all had a 50:50 distribution, but did all the models also have the same number of training samples?**

We added the following to the middle of the third paragraph of section 2.2:

*In the Umpqua, Calapooia, and Puerto Rico study sites, 68%, 28%, and 4% of the slope units contained landslides, respectively.*

We also added a table to the supplemental materials that provides the other requested statistics.

Table. S1 Number of landslide samples

| Location | Sampling Method | | | | |
|---|---|---|---|---|---|
| | 10m | 10m_med | 10_m_multi | 30m | SU |
| Umpqua | 3499 | 3499 | 3707 | 3090 | 1237 |
| Calapooia | 485 | 485 | 4824 | 484 | 1983 |
| Puerto Rico | NA | NA | NA | 71431 | 6263 |

Reference to this table is found at the end of the second paragraph of section 2.2 and reads as follows:

*Table S1 shows the number of points for each study site and sampling method, respectively.*

**Fig 5 and 6: What are the red and blue dots?**

We inserted the following into the caption of figure 5:

*the red and blue dots show the data outlying the whiskers;*

Reviewer #2 Comments
* * *
**I believe the manuscript is substantially improved with respect to the initial submission, the authors followed both reviewers' suggestions, and removing the detailed comparison with one existing model is probably a good thing. I still believe that the slope unit maps produced using the proposed software and the other one - in one of the removed figures - are not equivalent at all. This is crucial if one is to discuss the processing time of two software programs: if the output is different, or substantially different, the comparison is not justified. If the differences have an impact on landslide susceptibility is a different question, of course. I could further discuss about the issues of optimization, optimization algorithms, and the advantages of parametrization, but this is not the venue for that: I believe the manuscript deserves publication at this stage. One only suggestion, about one of the paragraphs in the supplementary material, also quoted in the authors' reply. The paragraph discussing the role of intermediate watersheds and the possibility of providing custom watersheds belongs to the main text, in my opinion. I am referring to the paragraph starting with "Creating intermediate watersheds allows ..." and ending with "each intermediate watershed in parallel."**

We thank the reviewer for their additional comments on our revised manuscript. The comparison to between r.slopeunits and SUMak was removed from the manuscript due to the reviewers' initial suggestions. There is no direct comparison between our method and any other slope unit software in the manuscript.

We moved the description of intermediate watersheds to section 2.1 in the main text. The new paragraph reads as follows:

*If the domain of interest has significant variation in topography, TauDEM may choose a threshold that doesn't adequately characterize every area within the domain. Thus, SUMak provides different options for subdividing the domain in preparation for the application of the slope unit optimization procedure described above. We refer to these preliminary subdivisions as intermediate watersheds. Intermediate watersheds must be small enough to limit the variation in topography but large enough to avoid significantly reducing computational efficiency. While experimenting with different watershed dimensions on the topographically diverse regions of Sicily, Puerto Rico, and the Umpqua and Calapooia watersheds, we found an accumulation threshold of ~100 km$^2$ to adequately strike this balance. This threshold can be adjusted to meet the user's needs, or SUMak has an option to input predetermined intermediate watersheds. After appropriate intermediate watersheds are created, the algorithm runs the rest of the processing steps individually for each intermediate watershed in parallel as detailed in Text S1 and the online repository (Woodard, 2023).*